# Two-sample survival tests based on control arm summary statistics

**Jannik Feld**[1]*, **Moritz Fabian Danzer**[1], **Andreas Faldum**[1], **Anastasia Janina Hobbach**[2], **Rene Schmidt**[1]

**1** Institute of Biostatistics and Clinical Research, University of Münster, Münster, Germany, **2** Department of Cardiology I, Coronary, Peripheral Vascular Disease and Heart Failure, University Hospital Münster, Münster, Germany

* Jannik.Feld@ukmuenster.de

**Data Availability Statement:** The authors confirm that the source of data utilized in the real world example cannot be made available in the manuscript, the supplemental files, or in a public repository due to German data protection laws

## Abstract

The one-sample log-rank test is the preferred method for analysing the outcome of single-arm survival trials. It compares the survival distribution of patients with a prefixed reference survival curve that usually represents the expected outcome under standard of care. However, classical one-sample log-rank tests assume that the reference curve is known, ignoring that it is frequently estimated from historical data and therefore susceptible to sampling error. Neglecting the variability of the reference curve can lead to an inflated type I error rate, as shown in a previous paper. Here, we propose a new survival test that allows to account for the sampling error of the reference curve without knowledge of the full underlying historical survival time data. Our new test allows to perform a valid historical comparison of patient survival times when only a historical survival curve rather than the full historic data is available. It thus applies in settings where the two-sample log-rank test is not applicable as method of choice due to non-availability of historic individual patient survival time data. We develop sample size calculation formulas, give an example application and study the performance of the new test in a simulation study.

## Introduction

The one-sample log-rank test [1] is the method of choice for single-arm Phase II trials with time-to-event endpoint. It allows to compare the survival of the patients to a prefixed reference survival curve that typically represents the expected survival under standard of care.

A central point of criticism of the one-sample log-rank test relates to the process of selecting the reference survival curve. It is common practice to choose the reference survival curve in the light of historic data on standard treatment. This implies that choice of the reference survival curve itself is prone to statistical error, which however, is ignored in the classical one-sample log-rank statistic. In classical log-rank tests the prefixed reference curve is rather treated as deterministic. Doing so results in an inflation of the type I error rate, when in fact the prefixed reference curve resulted from an estimation process [2].

For this reason, it is standard practice to use the two-sample log-rank test to compare patient survival with a historical control group whenever historical survival data are available.

('Bundesdatenschutzgesetz', BDSG). Therefore, they are stored on a secure drive in the WIdO, to facilitate replication of the results. Generally, access to data of statutory health insurance funds for research purposes is possible only under the conditions defined in German Social Law (SGB V § 287). Requests for data access can be sent as a formal proposal specifying the recipient and purpose of the data transfer to the appropriate data protection agency. Access to the data used in this study can only be provided to external parties under the conditions of the cooperation contract of this research project and after written approval by the sickness fund. For assistance in obtaining access to the data, please contact wido@wido.bv. aok.de.

**Funding:** The work of MFD was funded by the German Science Foundation (Deutsche Forschungsgemeinschaft, DFG, https://www.dfg. de, grant number 413730122). The study was conducted within the framework of the GenderVasc project (Gender-specific real care situation of patients with arteriosclerotic cardiovascular diseases) funded by The Federal Joint Committee, Innovation Committee (G-BA, Innovationsfond, number 01VSF18051). GenderVasc is a cooperation project with the AOK Research Institute (WIdO). The funders had no role in study design, data collection and analysis, decision to publish, or preparation of the manuscript.

**Competing interests:** The authors have declared that no competing interests exist.

However, situations may arise in which the full historical survival data on individual patient level is not (or no longer) available to the researcher. This may occur for a variety of reasons, ranging from copyright to privacy reasons, such as when data sharing is not covered by informed consent or individual patients withdrew their consent. In such a situation, generally only summary statistics such as a survival distribution estimator based on historical survival times together with basic data on sample size, event numbers, and length of follow-up will be available in peer-reviewed scientific publications. Then, application of the two-sample log-rank test is not possible because the required complete historical survival data on individual patient level are not available. Conversely, it is also not permissible to use the published survival estimator as deterministic reference curve in a classical one-sample log-rank test, since this ignores the sampling error of the reference curve, leading to inflation of the type I error rate [2].

In this paper, we present a solution to this problem. We propose a new survival test that, under mild assumptions, allows the sampling error of the reference curve to be taken into account without knowledge of the full historical survival time data. This allows to perform a valid historical comparison of patient survival times when only an estimator of the historical survival curve is available instead of the full historical survival data. Our proposed test can formally also be interpreted as a two-sample test for survival times, and can thus be compared with the two-sample log-rank test in terms of test performance. However, the intended application focus of our new test is comparison of survival against a historical reference in the absence of the full historical survival time data.

The paper is organized as follows. After settling notation and the testing problem, we describe the test statistic and its distributional properties. We then will provide an example application on real world data. Additionally, we provide sample size calculation methodology. Calculation of rejection regions and sample size are based on the approximate distribution of the new test statistic in the large sample limit. Therefore small sample properties of the new test regarding type I and type II error rate control are studied by simulation, and compared to the two-sample log-rank test. We conclude with a discussion of future research.

## General aspects

### Notation

We consider a single-arm clinical trial with survival data from a treatment group $B$, say (experimental intervention, prospectively collected data). This survival data shall be tested against the survival under standard of care treatment $A$, say represented by the reference survival curve $S_{T_A}$. However, in practice the true survival curve under standard of care $S_{T_A}$ is unknown and thus researchers may rely on an estimator of the reference survival curve $\widehat{S}_{T_A}$ taken e.g. from a publication in a peer reviewed journal. We denote with $\widehat{\Lambda}_A$ a corresponding estimator of the cumulative hazard either taken from the literature or obtained through the equation $\widehat{\Lambda}_A = -\log(\widehat{S}_{T_A})$. These estimators stem from a group of historical patients who were treated under standard of care. However, the individual data on patient level are usually not available for the public due to data protection and privacy reasons.

Let $\mathcal{N}_B$ denote the set of patients from group $B$, $n_B | \mathcal{N}_B := |$ the number of such patients. We denote by $T_{B,i}$ and $C_{B,i}$ the time from entry to event or censoring for patient $i \in \mathcal{N}_B$ respectively. Let $X_{B,i} := T_{B,i} \wedge C_{B,i}$ denote the minimum of both. As usual, we assume that the $T_{B,i}$ and $C_{B,i}$ are mutually independent (non-informative censoring).

Based on the observed data, we calculate the *number of events* from treatment group $B$ up to study time $s \geq 0$ as

$$N_B(s) := \sum_{i \in \mathcal{N}_B} I(T_{B,i} \leq s, T_{B,i} \leq C_{B,i}). \tag{1}$$

As usual, we let $\lambda_B(s) := \lim_{\Delta \to 0} P(s \leq T_{B,i} < s + \Delta | T_{B,i} \geq s)/\Delta$ denote the true hazard of a patient $i \in \mathcal{N}_B$ and denote by $\Lambda_B(s) := \int_0^s \lambda_B(u) \, du$ the corresponding cumulative hazard function for $s \geq 0$, respectively.

Finally, we denote by $F_{T_B}$ and $S_{T_B}$ the distribution function and survival function of the time to event $T_{B,i}$ and denote with $f_{C_B}$ the density function of the censoring time $C_{B,i}$ for $i \in \mathcal{N}_B$. We assume the tuples $(C_{B,i}, T_{B,i})_i$ to be independent and identically distributed and assume $C_{B,i}$ and $T_{B,i}$ to be independent for each $i \in \mathcal{N}_B$. We denote by $a_B$ the recruitment period length, with $f_B$ the follow up period length resulting in a trial period length of $a_B + f_B$ for the new intervention trial. In particular, we assume that patients enter the trial between calendar time 0 and $a_B$, and are then further followed-up until calendar time $a_B + f_B$, unless they are not censored or experience an event. Therefore the intended follow up of the first patient is $a_B + f_B$ and of the last patient is $f_B$ respectively. Furthermore we denote with $r_B := n_B/a_B$ the recruitment rate of the new trial.

The above definitions shall be applied analogously to group A. In particular, we denote by $\pi = n_B/n_A$ the ratio of the group sizes. However, notice again that usually the individual data $X_{A,i}$, $T_{A,i}$ and $C_{A,i}$ for $i \in \mathcal{N}_A$ are unknown to the researcher and only the survival curve estimator $\widehat{S}_{T_A}$ with summary statistics about number at risk and sample size are available.

## The testing procedure

We are interested in testing

$$H_0 : \Lambda_B(s) = \Lambda_A(s) \text{ for all } s \in [0, s_{max}],$$

for some prefixed time-horizon $s_{max} > 0$, i.e. that the survival of the patients under the new experimental treatment coincides with survival under standard of care. The classical one-sample log-rank test could be used to test $H_0$ if the reference cumulative hazard function $\Lambda_A$ was known. However, the practical user faces the problem that the true reference curve $\Lambda_A$ representing survival under standard of care is unknown. For this reason $\Lambda_A$ is often replaced within all test statistics by its Nelson-Aalen estimate $\widehat{\Lambda}_A$ derived from historic data. This means that the classical one-sample log-rank test of $H_0$ is based on the statistic

$$Z_{\text{OSLR}} := \frac{\widehat{M}_0(s_{\max})}{\widehat{\Sigma}_{\text{OSLR}}(s_{\max})}, \tag{2}$$

with

$$\widehat{M}_0(s) := n_B^{-1/2}[N_B(s) - \sum_{i \in \mathcal{N}_B} \widehat{\Lambda}_A(s \wedge X_{B,i})] \tag{3}$$

and estimator of $\text{Var}(\widehat{M}_0(s))$

$$\widehat{\Sigma}_{\text{OSLR}}^2(s) := 1/2 \cdot n_B^{-1}[N_B(s) + \sum_{i \in \mathcal{N}_B} \widehat{\Lambda}_A(s \wedge X_{B,i})]$$

rejecting $H_0$ whenever $|Z_{\text{OSLR}}| \geq \Phi^{-1}(1 - \alpha/2)$, where $\Phi^{-1}$ is the standard normal quantile

function. This proceeding would only be admissible if the variance of $\widehat{\Lambda}_A$ was negligible. In practice, however, $\mathrm{Var}(\widehat{\Lambda}_A)$ is typically not negligible [2]. Then, using the process $\widehat{M}_0$ with the classical standardisation $\widehat{\Sigma}^2_{\mathrm{OSLR}}$ leads to an under-estimation of variance and thus inflation of the type I error rate. The factor of under-estimation of variance may be approximated by

$$R^2 \approx 1/(1 + \pi) \qquad \text{(see Eq. (13) and Appendix B in [2]),}$$

if recruitment and censoring mechanism were equal in the new treatment group B and the historical group underlying the estimate $\widehat{\Lambda}_A$. This motivates us to propose the following random variable as a new survival test statistic

$$Z_\pi := \frac{\widehat{M}_0(s_{\max})}{\widehat{\Sigma}_{\mathrm{OSLR}}(s_{\max}) \cdot \sqrt{1 + \pi}}. \tag{4}$$

Then, an approximate level $\alpha$ test of $H_0$ is defined by rejecting $H_0$ whenever

$$|Z_\pi| \geq \Phi^{-1}(1 - \alpha/2), \tag{5}$$

where $\alpha$ is the desired two-sided significance level. Notice again that we used the assumption of equal recruitment and censoring mechanisms in the new experimental and historic control cohort. We will keep this assumption for the rest of the manuscript and will investigate how deviation from this assumption affects the type I error rate control within simulation studies. Furthermore, notice that procedure (5) is equivalent to adjusting the nominal $\alpha$-level of the classical one-sample log-rank test methodology to

$$\alpha_{\mathrm{nominal}} := 2 \cdot \Phi(z_{\alpha/2} \cdot \sqrt{1 + \pi})$$

or analogously using the critical boundary

$$z_{1 - \alpha_{\mathrm{nominal}}/2} := z_{1 - \alpha/2} \cdot \sqrt{1 + \pi}.$$

Notice that in theory one could use a corrected variance estimator $\widehat{\Sigma}^2$ for $\mathrm{Var}(\widehat{M}_0(s))$ (see Eq. (5) of [2]) instead of our approximation $\widehat{\Sigma}^2_{\mathrm{OSLR}}(s_{\max}) \cdot (1 + \pi)$. However, using the statistic $Z_\pi$ yields some advantages. Firstly, calculation of $Z_\pi$ can be executed with existing standard software whereas $\widehat{\Sigma}^2$ needs additional implementation effort to be calculated. Secondly, for computation of $\widehat{\Sigma}^2$ the full dataset of the historical patients is needed, whereas $Z_\pi$ is based only on the sample size $n_A$ and the estimated reference curve $\widehat{\Lambda}_A$ [3] (or equivalently the Kaplan Meier estimator $\widehat{S}_A$ [4]). This enables the usage of the $Z_\pi$-test (5) even in absence of full historical survival time data.

## A real world example

Our hypothetical example stems from clinical cardiology and originates from the context of the 'obesity paradox'. This paradox refers to the observation that in certain cardiovascular diseases, overweight patients may exhibit a more favourable prognosis compared to non-obese patients. This phenomenon has been observed in several cardiovascular diseases, including heart failure, coronary heart disease and acute myocardial infarction. Despite the well-established link between obesity and an increased risk of cardiovascular disease, studies have found that obese individuals particularly those with pre-existing cardiovascular disease, may have a lower mortality risk compared to non-obese patients. This observation has generated considerable interest and debate in the medical community. In this context it might be interesting to

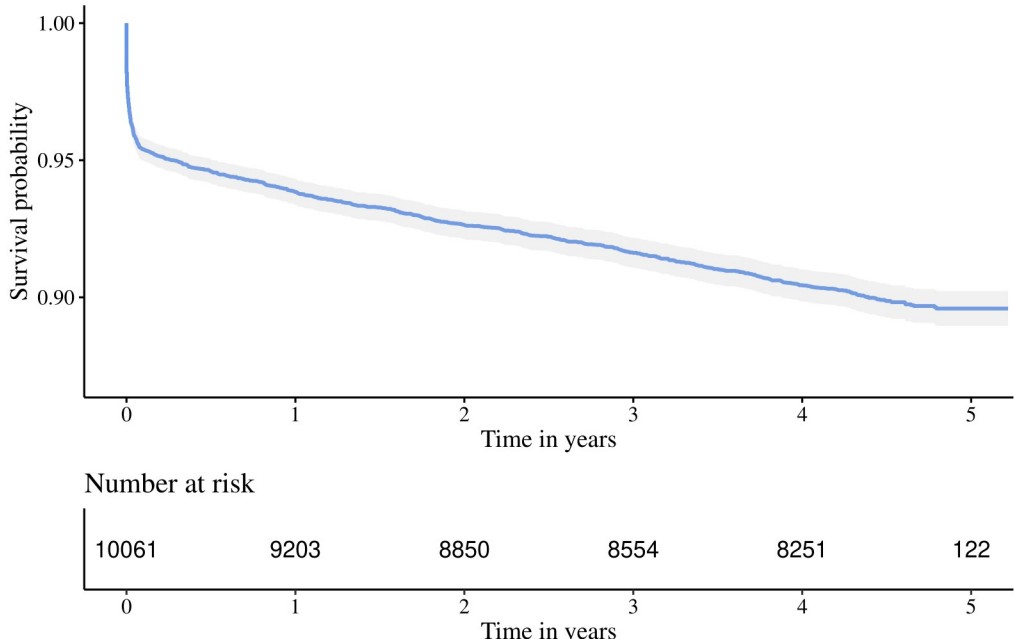

**Fig 1. Reference survival curve.** Kaplan Meier estimator $\widehat{S}_{T_A}$ of overall survival of patients hospitalized with a myocardial infarction in 2014 aged 30–60, without obesity and without a prior indication of chronic heart failure.

test the null hypothesis $H_0$ that the survival of obese patients suffering from acute myocardial infarction coincides with the survival of non-obese patients suffering acute myocardial infarction against a significance level of $\alpha = 5\%$.

For researchers it could be beneficial to use reference survival curves published by large registries or obtained from health insurance data, as these could serve as a suitable option for representing the general population.

However, in our experience, when health insurance companies grant researchers access to their insurance data they strictly prohibit to extract individual patient level data. Thus our method to make a comparison based upon summary statistics is needed. The following survival curve in Fig 1 is obtained from data of the "AOK- Die Gesundheitskasse" (AOK), a collective of 11 local German health insurance funds covering one third of the German population.

It includes all patients hospitalized with a myocardial infarction in 2014 aged 30–60, without obesity and without a prior indication of chronic heart failure. For our hypothetical example, this survival curve should take on the role of the historically estimated reference curve $\widehat{S}_{T_A}$ on which the access to the underlying patient level data $X_{A,i}, I(T_{A,i} \leq C_{A,i})$ for $i \in \mathcal{N}_A$ may be restricted to the researchers.

We used the WebPlotDigitizer website (https://apps.automeris.io/wpd/) to extract the x-y data at 481 points from the obtained image, which we interpolated to reconstruct the reference survival function $\widehat{S}_{T_A}$ to finally calculate the reference cumulative hazard function

$$\widehat{\Lambda}_A = -\log(\widehat{S}_{T_A}).$$

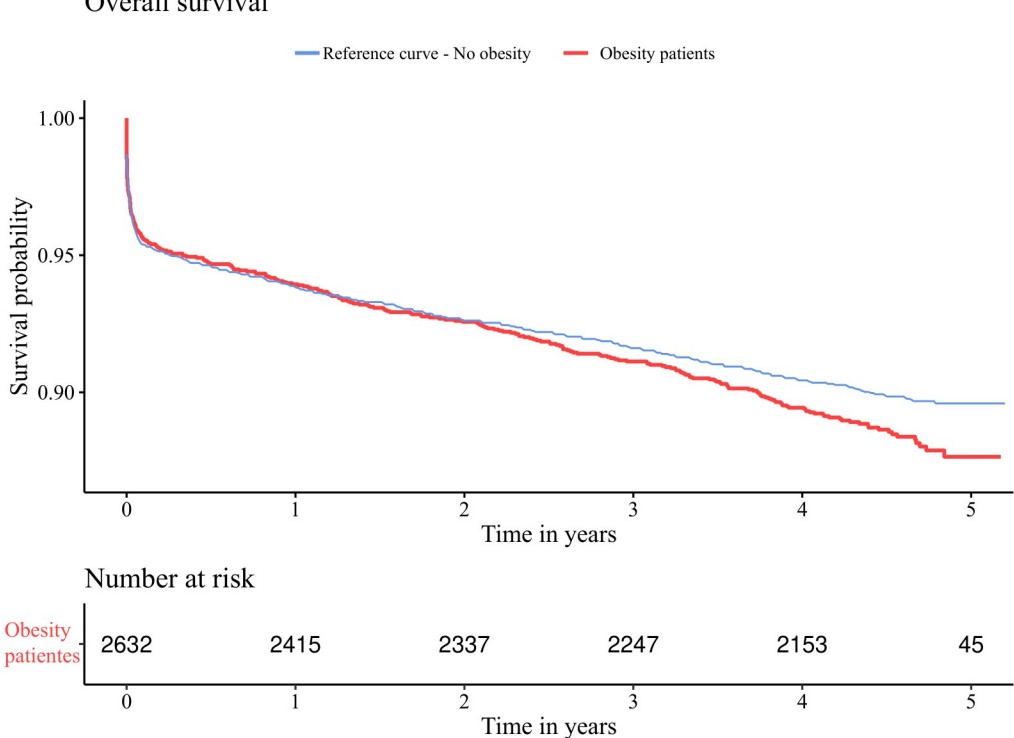

**Fig 2. Overall survival of both collectives.** Kaplan Meier estimators $\widehat{S}_{T_A}$ and $\widehat{S}_{T_B}$ of overall survival of patients hospitalized with a myocardial infarction in 2014 aged 30–60, without a prior indication of chronic heart failure.

We used this reference cumulative hazard function to test against a sample of obese patients from the year 2014 aged 30–60 without a prior indication of chronic heart failure representing the set of patients $\mathcal{N}_B$. The Kaplan Meier estimators $\widehat{S}_{T_A}$ and $\widehat{S}_{T_B}$ of both collectives are presented in Fig 2.

If the classical one-sample log-rank test (2) is used without considering the sampling variability of the reference curve, the test would yield a p-value of $p_{\text{OSLR}} = 0.0320$, yielding a significant difference between the two groups. However using our new technique (4) with $\pi = 2632/10061$ we obtain a p-value of $p_{\pi} = 0.0562$ being non significant.

Another approach to test the two groups would be to use a reconstruction algorithm [5] to obtain reconstructed censored survival times $\tilde{X}_{A,i}$ for $i \in \mathcal{N}_A$ on individual patient level of the reference cohort to plug into the classical two-sample log-rank test. This approach would lead to a p-value of $p_{\text{TSR}} = 0.0449$, yielding statistical significance once again. However, this procedure can only be used with great caution. On one hand, reconstruction algorithms deliver solid results when used to reproduce Kaplan Meier point estimators since they are designed to reproduce these as accurately as possible. However, using the reconstructed data for estimating hazard ratios or plugging them into a two-sample log-rank test may lead to poor results, as small, but systemic, deviations that are of little relevance for point estimators may add up over the course of the tested time frame [5]. These deviations are not accounted for by the methodology of the two-sample log-rank test and may endanger the correct type I error rate control. This effect is especially prominent when the number of extracted x-y data points from the Kaplan Meier plot image is lower than the number of individual patients data points which need to be reconstructed. In this case the algorithm is under-saturated.

Finally, if researchers had both original data sets and applied a two-sample log-rank test, the resulting p-value would be $p_{TS} = 0.0503$ indicating no significance. Thus, both the classical one-sample log-rank test and the two-sample log-rank test using reconstructed data resulted in anti-conservative p-values. In summary our analysis agrees with the original two-sample log-rank test which does not confirm the obesity paradox for the cohort of patients aged 30–60 suffering from acute myocardial infarction.

## Sample size calculation

### General formula

Testing $H_0$ to the level $\alpha$ using $Z_\pi$ is equivalent to testing $H_0$ using the classical one-sample log-rank test to the adjusted level $\alpha_{\text{nominal}}$. Thus the sample size formula for the classical one-sample log-rank test from [6] immediately yields a sample size formula for the new test based on $Z_\pi$. In order to achieve a power of approximately $1 - \beta$ for an allocated significance level $\alpha$ and constant hazard ratio $\omega := \Lambda_B(s)/\Lambda_A(s)$ for all $s \geq 0$, the new test procedure (5) has to be performed based on a total of

$$d = \omega \left( \frac{\Phi^{-1}(1 - \alpha/2) \cdot \sqrt{1 + \pi} + \sqrt{\omega} \cdot \Phi^{-1}(1 - \beta)}{1 - \omega} \right)^2$$

events [6]. Notice that above equation is a direct result of plugging our adjusted significance level $\alpha_{\text{nominal}}$ into the sample size formula of the classical one-sample log-rank test (see Eq. (4) of [6]). However the existing methodology and software can not be used directly as the adjusted significance level $\alpha_{\text{nominal}}$ itself depends on the sample size through $\pi$. This number of events $d$ can be expected to be achieved if the sample size is chosen such that

$$d = E[N_B(s_{\max})] = n_B \cdot \int_0^\infty F_{T_B}(s_{\max} \wedge u) f_{C_B}(u) \ du.$$

Through administrative censoring at calendar time $a_B + f_B$ it is clear that $f_{C_B}$ is only non-zero on the interval $[0, a_B + f_B]$ and thus the integral limits reduce accordingly. Through fixing two of the three parameters $a_B$, $f_B$ and $r_B$ by clinical consideration and by using the identities $n_B = r_B \cdot a_B$ and $\pi = (r_B \cdot a_B)/n_A$ as well as recognizing the functional dependency of $f_{C_B}$ on the parameters $a_B$ and $f_B$, one can numerically solve the above equation for the last free parameter of $a_B$, $f_B$ and $r_B$. We provide R-Code in the supplemental material to perform this sample size calculation.

### Closed formula for rare late events

The above methodology describes a general approach for sample size calculation. However, in the clinical setting, it may be advantageous for the statistician to have a closed formula at hand. For this reason we will develop a closed formula for a special case often met in practice.

We assume the following conditions:

(*i*): No censoring except for administrative censoring at calendar time $a_B + f_B$

(*ii*): A plateau of the survival function at the edge of the observation window $[0, s_{\max}]$ such that $1 - S_{T_A}(s_{\max}) \approx 1 - S_{T_A}(f_B)$

Assumption (*ii*) may be approximately met when the survival curve stagnates for a long period of time. An example might be given by the survival curve of patients after a complex

surgery, which sharply declines shortly after the procedure but stagnates after a year. Also, in the context of cause specific deaths such a plateau may occur naturally.

We now evaluate the power formula

$$E[N_B(s_{\max})] = \omega \cdot \left( \frac{\Phi^{-1}(1-\alpha/2) \cdot \sqrt{1+\pi} + \sqrt{\omega} \cdot \Phi^{-1}(1-\beta)}{1-\omega} \right)^2. \qquad (6)$$

Under the assumptions (*i*) and (*ii*), the probability of observing an event is equal to the probability of having an event in general, and thus the expectation on the left hand side is tantamount to $n_B \cdot p_B$, with $p_B := 1 - S_{T_A}(s_{\max})^\omega$. We can obtain the required sample size $n_B$ by solving the above equation for for $\zeta := \sqrt{1 + \frac{n_B}{n_A}}$. Furthermore, using the identity $\pi = n_B/n_A$, solving the equation above is equivalent to calculating the roots of the quadratic polynomial

$$[n_A \cdot p_B \cdot (1-\omega)^2 - \omega \cdot z_{1-\alpha/2}^2] \cdot \zeta^2 + [-2 \cdot \omega^{3/2} \cdot z_{1-\alpha/2} \cdot z_{1-\beta}] \cdot \zeta + [-\omega^2 \cdot z_{1-\beta}^2 - n_A \cdot p_B \cdot (1-\omega)^2].$$

Notice that since the intercept and linear coefficient are negative for $1 - \beta > 0.5$, the polynomial has a positive root if and only if the quadratic coefficient is positive e.g.

$$n_A > \frac{\omega \cdot z_{1-\alpha/2}^2}{p_B \cdot (1-\omega)^2} \qquad \text{if } 1 - \beta > 0.5. \qquad (7)$$

This yields a lower bound for $n_A$. When $n_A$ falls below this threshold it is not possible to get a significant result to a power of at least 50% when sampling variability is taken into account. Notice that the assumptions (*i*) and (*ii*) were introduced, to easily approximate the probability of observing an event $P(T_B < C_B)$. However, using the inequality $P(T_B < C_B) < 1 - S_{T_A}(a_B + f_B)^\omega$, we can also obtain a lower bound for the more general sample size calculation:

$$n_A > \frac{\omega \cdot z_{1-\alpha/2}^2}{(1 - S_{T_A}(a_B + f_B)^\omega) \cdot (1-\omega)^2} \qquad \text{if } 1 - \beta > 0.5. \qquad (8)$$

Furthermore if inequality (7) holds, the distinct positive root of the polynomial can be calculated and thus the required sample size $n_B$ is given by

$$n_B = n_A \cdot \left( \frac{-c_1 + \sqrt{c_1^2 - 4c_2 c_0}}{2c_2} \right)^2 - n_A,$$

where
$c_2 := n_A \cdot p_A{}^\omega \cdot (1-\omega)^2 - \omega \cdot z_{1-\alpha/2}^2$ is the quadratic coefficient,
$c_1 := -2 \cdot \omega^{3/2} \cdot z_{1-\alpha/2} \cdot z_{1-\beta}$ is the linear coefficient and
$c_0 := -\omega^2 \cdot z_{1-\beta}^2 - n_A \cdot p_A{}^\omega \cdot (1-\omega)^2$ is the intercept.

## Simulation study: Comparison with the two-sample log-rank test

### Design

We proposed a significance test for the null hypothesis $H_0$ based on the approximate large sample distribution of the test statistic $Z_\pi$ introduced before. Notice that only the sample size $n_A$ and the graph of the Nelson Aalen estimate $\widehat{\Lambda}_A$ of group $A$ is needed to compute the test statistic $Z_\pi$. Knowledge of full survival data for group $A$ is not required. However, in case of full knowledge

of the data of group $A$, our test with consideration of reference curve variability, may also be interpreted as a two-sample survival test. This simulation, therefore aims to study performance of the new survival test for sample sizes of practical relevance, compared to the classical two-sample log-rank test, which by design takes the variability of the historical data into account (see [7, 8]). Asymptotically (i.e. for sufficiently large sample size) the classical two-sample log-rank test is known to be the optimal two-sample test under proportional hazards (PH) alternatives. It is thus of particular interest to compare performance of both tests under PH alternatives.

In clinical practice the sample size calculation methodology is most likely to be used with the recruitment rate $r_B$ and follow up duration $f_B$ fixed by clinical consideration and thus solving for the accrual period length $a_B$, resulting in the sample size $n_B \coloneqq r_B \cdot a_B$. However, since our approximation is based on the assumption that the censoring and recruitment mechanisms are equal to the historic cohort, we use the sample size equation to solve for $r_B$ with predefined accrual period length $a_B$ and follow up duration $f_B$ (and thus fixing the administrative censoring and recruitment mechanism independently from the sample size), to have clear insight in the performance of our test if the assumption of equal censoring and recruitment mechanisms is violated.

In our simulations, patients were assumed to enter the trial uniformly between year 0 and year $a_A = a_B = 3$ and followed up for $f_B \in \{2, 3, 4\}$ in the intervention cohort and $f_A = 3$ years in the historic cohort.

Accordingly, the calendar times of entry were generated according to a uniform distribution $E_{x,i} \sim \mathcal{U}(0, a_x)$ on $[0, a_x]$.

We considered scenarios with no loss to follow-up setting $C_{x,i} \coloneqq a + f_x - E_{x,i} \sim \mathcal{U}(f_x, a + f_x)$ and also considered scenarios with additional independent exponentially distributed censoring $C^*_{x,i} \sim \text{Exp}(\lambda_{C_x})$, where overall censoring was set as $C_{x,i} \coloneqq (a + f_x - E_{x,i}) \wedge C^*_{x,i}$. In these Scenarios we set $\lambda_{C_A} = 0.20$ and varied $\lambda_{C_B} \in \{0.15, 0.20, 0.25\}$ between scenarios to analyse the robustness of the test, when censoring mechanisms are not equal.

Survival times $T_{A,i}$ in the control group $A$ were generated according to a Weibull distribution i.e. $\Lambda_A(s) \coloneqq -\log(s) \cdot t^\kappa$ with prefixed shape parameter $\kappa \in \{0.1, 0.25, 0.5, 1.0, 1.5, 2.0, 5.0\}$ and 1-year survival rate $S_1 \coloneqq S_{T_A}(1) \coloneqq 0.5$. To implement the PH condition, survival times $T_{B,i}$ in the experimental intervention group $B$ were generated according to a Weibull distribution i.e. $\Lambda_B(s) 2/3 \cdot \Lambda_A(s)$, fixing $\omega = 2/3$ as the true hazard ratio such that $S_{T_B}(1) \coloneqq 0.5^{2/3} \approx 0.63$. Additional simulation results with $\omega = 1/2$ and $\omega = 4/5$ will be presented in the S1 Appendix.

We let the number of patients in the historic group $n_A \in \{200, 400, 800\}$ vary between scenarios and used our sample size equation to determine the required number of patients in the intervention group $n_B$ for a targeted power of $1 - \beta = 80\%$ for our new test under planning alternative $K_0: \Lambda_B = 2/3 \cdot \Lambda_A$ for a two-sided significance level of $\alpha = 5\%$. From these parameter constellations we omitted any scenario with $\pi > 2$ since these might be unrealistic in practical use of the methodology.

For each parameter constellation, we generated 10000 samples to which we applied both the new test as well as the classical two-sample log-rank test. Reported in Tables 1 and 2 are the empirical type I and type II error rates for each parameter constellation. To increase readability of the tables we omitted potential confidence intervals for our estimators, but notice that with 10000 samples potential 95%-confidence intervals for the type I error rate estimators given in Tables 1 and 2 should have a width of about 0.004.

## Results

Reassuringly, our new test statistic keeps the significance level of $\alpha = 5\%$. Even in scenarios where the equal censoring mechanism assumption was moderately violated, the error rate

**Table 1. Comparison of empirical type I and II errors of the new procedure and two-sample log-rank test with varying follow-up length.**

| $\kappa$ | $n_A$ | $f_B = 2$ | | | | | $f_B = 3$ | | | | | $f_B = 4$ | | | | |
|---|---|---|---|---|---|---|---|---|---|---|---|---|---|---|---|---|
| | | $n_B$ | $\alpha_{TS}$ | $\alpha_\pi$ | $1 - \beta_{TS}$ | $1 - \beta_\pi$ | $n_B$ | $\alpha_{TS}$ | $\alpha_\pi$ | $1 - \beta_{TS}$ | $1 - \beta_\pi$ | $n_B$ | $\alpha_{TS}$ | $\alpha_\pi$ | $1 - \beta_{TS}$ | $1 - \beta_\pi$ |
| 0.10 | 200 | 164 | 0.051 | 0.051 | 0.760 | 0.761 | 159 | 0.048 | 0.049 | 0.758 | 0.759 | 155 | 0.052 | 0.052 | 0.743 | 0.751 |
| | 400 | 128 | 0.048 | 0.048 | 0.803 | 0.787 | 125 | 0.050 | 0.050 | 0.813 | 0.796 | 122 | 0.050 | 0.050 | 0.768 | 0.791 |
| | 800 | 115 | 0.052 | 0.049 | 0.836 | 0.811 | 112 | 0.052 | 0.051 | 0.830 | 0.809 | 110 | 0.052 | 0.051 | 0.773 | 0.805 |
| 0.25 | 200 | 134 | 0.047 | 0.047 | 0.767 | 0.761 | 125 | 0.049 | 0.049 | 0.767 | 0.765 | 119 | 0.055 | 0.058 | 0.740 | 0.747 |
| | 400 | 108 | 0.047 | 0.047 | 0.806 | 0.786 | 103 | 0.052 | 0.052 | 0.810 | 0.792 | 98 | 0.051 | 0.051 | 0.760 | 0.783 |
| | 800 | 99 | 0.051 | 0.050 | 0.830 | 0.802 | 94 | 0.052 | 0.051 | 0.832 | 0.807 | 90 | 0.051 | 0.050 | 0.762 | 0.790 |
| 0.50 | 200 | 100 | 0.047 | 0.047 | 0.778 | 0.768 | 90 | 0.047 | 0.047 | 0.777 | 0.769 | 83 | 0.051 | 0.052 | 0.727 | 0.741 |
| | 400 | 85 | 0.050 | 0.048 | 0.809 | 0.787 | 78 | 0.049 | 0.049 | 0.810 | 0.791 | 73 | 0.053 | 0.051 | 0.755 | 0.774 |
| | 800 | 79 | 0.050 | 0.050 | 0.834 | 0.806 | 73 | 0.049 | 0.048 | 0.820 | 0.795 | 68 | 0.052 | 0.050 | 0.758 | 0.782 |
| 1.00 | 200 | 67 | 0.045 | 0.045 | 0.783 | 0.766 | 60 | 0.051 | 0.052 | 0.775 | 0.758 | 56 | 0.051 | 0.046 | 0.708 | 0.721 |
| | 400 | 60 | 0.049 | 0.049 | 0.809 | 0.786 | 54 | 0.051 | 0.050 | 0.793 | 0.771 | 51 | 0.053 | 0.048 | 0.737 | 0.763 |
| | 800 | 57 | 0.050 | 0.049 | 0.816 | 0.790 | 51 | 0.052 | 0.052 | 0.803 | 0.775 | 48 | 0.049 | 0.046 | 0.737 | 0.764 |
| 1.50 | 200 | 55 | 0.051 | 0.050 | 0.769 | 0.746 | 52 | 0.044 | 0.043 | 0.749 | 0.716 | 50 | 0.056 | 0.048 | 0.698 | 0.700 |
| | 400 | 50 | 0.049 | 0.050 | 0.786 | 0.760 | 47 | 0.048 | 0.046 | 0.782 | 0.749 | 46 | 0.054 | 0.049 | 0.722 | 0.742 |
| | 800 | 48 | 0.051 | 0.051 | 0.800 | 0.770 | 45 | 0.051 | 0.051 | 0.792 | 0.758 | 44 | 0.055 | 0.051 | 0.723 | 0.748 |
| 2.00 | 200 | 52 | 0.047 | 0.045 | 0.762 | 0.732 | 50 | 0.049 | 0.046 | 0.755 | 0.714 | 50 | 0.056 | 0.046 | 0.703 | 0.707 |
| | 400 | 47 | 0.051 | 0.050 | 0.780 | 0.746 | 46 | 0.051 | 0.050 | 0.779 | 0.741 | 46 | 0.052 | 0.045 | 0.715 | 0.734 |
| | 800 | 45 | 0.048 | 0.048 | 0.795 | 0.763 | 44 | 0.051 | 0.051 | 0.783 | 0.746 | 44 | 0.049 | 0.044 | 0.724 | 0.751 |
| 5.00 | 200 | 50 | 0.051 | 0.048 | 0.758 | 0.718 | 50 | 0.052 | 0.048 | 0.755 | 0.709 | 50 | 0.058 | 0.050 | 0.702 | 0.705 |
| | 400 | 46 | 0.049 | 0.046 | 0.776 | 0.737 | 46 | 0.048 | 0.048 | 0.775 | 0.734 | 46 | 0.054 | 0.048 | 0.716 | 0.736 |
| | 800 | 44 | 0.049 | 0.048 | 0.790 | 0.751 | 44 | 0.047 | 0.045 | 0.788 | 0.752 | 44 | 0.052 | 0.048 | 0.730 | 0.757 |

Empirical type I error rate ($\alpha_\pi$ and $\alpha_{TS}$) and power ($1 - \beta_\pi$ and $1 - \beta_{TS}$) for the new test and for the classical two-sample log-rank test, respectively, under proportional hazards alternatives for Weibull distributed survival times with shape parameter $\kappa$ and 1-year survival rate $S_1 = 0.5$ in the control arm. Theoretical two-sided significance level: 5%. Underlying sample size of the historical group $n_A$ was predefined whereas the sample size of the intervention group $n_B$ was calculated to achieve a theoretical power of 80% under the planning alternative $H_1: \Lambda_B = 2/3 \cdot \Lambda_A$ for the new test statistic using the sample size methodology presented in the previous section. No censoring despite administrative censoring after $f_A = 3$ and $f_B$ years in the historical and intervention group respectively.

inflation was negligible. The performance of the new test is comparable to the two-sample log-rank test which is known to be asymptotically optimal under the proportional hazards assumption. In scenarios with early events (Weibull shape parameter $\leq 1.5$) the performance was even slightly better. The sample size calculation algorithm, however had slightly mixed results. Many scenarios achieved the targeted power of 80% quite accurately. However, in some extreme scenarios with particularly small sample sizes and extreme shape parameters of the reference survival-curve, the sample size calculation yielded underpowered trials. For these combinations, we recommend validating the sample size by simulation and comparing the sample size to a two-armed trial using the two-sample log-rank test.

## Discussion

Traditional one-sample log-rank tests compare the survival of an experimental treatment to a prefixed reference survival curve, which typically represents the expected survival under standard of care. The choice of the reference survival curve is often based on historic data on standard therapy and thus prone to statistical error. Nevertheless, traditional one-sample log-rank tests do not account for this variance of the reference curve estimator. This non-consideration of the sampling variability leads to an inflation of the type I error rate. The extent of this

**Table 2. Comparison of empirical type I and II errors of the new procedure and two-sample log-rank test with varying censoring mechanism.**

| $\kappa$ | $n_A$ | $\lambda_{C_B} = 0.15$ | | | | | $\lambda_{C_B} = 0.20$ | | | | | $\lambda_{C_B} = 0.25$ | | | | |
|---|---|---|---|---|---|---|---|---|---|---|---|---|---|---|---|---|
| | | $n_B$ | $\alpha_{TS}$ | $\alpha_\pi$ | $1-\beta_{TS}$ | $1-\beta_\pi$ | $n_B$ | $\alpha_{TS}$ | $\alpha_\pi$ | $1-\beta_{TS}$ | $1-\beta_\pi$ | $n_B$ | $\alpha_{TS}$ | $\alpha_\pi$ | $1-\beta_{TS}$ | $1-\beta_\pi$ |
| 0.10 | 200 | 161 | 0.048 | 0.050 | 0.741 | 0.754 | 162 | 0.051 | 0.053 | 0.741 | 0.756 | 163 | 0.050 | 0.053 | 0.742 | 0.753 |
| | 400 | 126 | 0.050 | 0.050 | 0.759 | 0.789 | 127 | 0.052 | 0.053 | 0.772 | 0.798 | 127 | 0.051 | 0.052 | 0.762 | 0.788 |
| | 800 | 113 | 0.047 | 0.048 | 0.774 | 0.808 | 114 | 0.046 | 0.047 | 0.775 | 0.809 | 114 | 0.048 | 0.051 | 0.776 | 0.810 |
| 0.25 | 200 | 129 | 0.050 | 0.055 | 0.727 | 0.752 | 130 | 0.050 | 0.055 | 0.729 | 0.754 | 131 | 0.051 | 0.056 | 0.734 | 0.757 |
| | 400 | 105 | 0.051 | 0.053 | 0.754 | 0.786 | 106 | 0.049 | 0.049 | 0.750 | 0.784 | 107 | 0.052 | 0.054 | 0.760 | 0.792 |
| | 800 | 96 | 0.050 | 0.052 | 0.759 | 0.794 | 97 | 0.047 | 0.048 | 0.766 | 0.800 | 98 | 0.048 | 0.051 | 0.778 | 0.811 |
| 0.50 | 200 | 94 | 0.053 | 0.059 | 0.717 | 0.751 | 96 | 0.053 | 0.059 | 0.719 | 0.755 | 97 | 0.049 | 0.057 | 0.720 | 0.753 |
| | 400 | 81 | 0.049 | 0.053 | 0.747 | 0.786 | 82 | 0.052 | 0.052 | 0.745 | 0.779 | 83 | 0.054 | 0.058 | 0.743 | 0.780 |
| | 800 | 75 | 0.057 | 0.057 | 0.756 | 0.794 | 76 | 0.051 | 0.053 | 0.759 | 0.800 | 77 | 0.049 | 0.049 | 0.761 | 0.800 |
| 1.00 | 200 | 63 | 0.050 | 0.054 | 0.698 | 0.725 | 64 | 0.051 | 0.054 | 0.702 | 0.734 | 65 | 0.050 | 0.050 | 0.703 | 0.732 |
| | 400 | 57 | 0.053 | 0.052 | 0.722 | 0.757 | 57 | 0.052 | 0.051 | 0.728 | 0.763 | 58 | 0.046 | 0.047 | 0.724 | 0.761 |
| | 800 | 54 | 0.049 | 0.049 | 0.740 | 0.778 | 55 | 0.052 | 0.050 | 0.742 | 0.780 | 56 | 0.051 | 0.050 | 0.744 | 0.780 |
| 1.50 | 200 | 54 | 0.055 | 0.052 | 0.688 | 0.699 | 55 | 0.052 | 0.050 | 0.692 | 0.704 | 56 | 0.057 | 0.053 | 0.693 | 0.703 |
| | 400 | 49 | 0.056 | 0.053 | 0.710 | 0.737 | 50 | 0.051 | 0.050 | 0.716 | 0.740 | 51 | 0.052 | 0.048 | 0.718 | 0.744 |
| | 800 | 47 | 0.054 | 0.050 | 0.720 | 0.749 | 48 | 0.052 | 0.048 | 0.723 | 0.754 | 49 | 0.052 | 0.048 | 0.728 | 0.758 |
| 2.00 | 200 | 53 | 0.054 | 0.048 | 0.689 | 0.693 | 53 | 0.050 | 0.046 | 0.687 | 0.691 | 54 | 0.052 | 0.048 | 0.693 | 0.695 |
| | 400 | 48 | 0.056 | 0.052 | 0.708 | 0.731 | 49 | 0.049 | 0.043 | 0.711 | 0.731 | 49 | 0.054 | 0.049 | 0.704 | 0.725 |
| | 800 | 46 | 0.050 | 0.045 | 0.721 | 0.749 | 47 | 0.056 | 0.051 | 0.715 | 0.741 | 47 | 0.056 | 0.052 | 0.718 | 0.748 |
| 5.00 | 200 | 52 | 0.051 | 0.047 | 0.695 | 0.699 | 53 | 0.054 | 0.050 | 0.691 | 0.698 | 53 | 0.051 | 0.046 | 0.683 | 0.686 |
| | 400 | 48 | 0.053 | 0.050 | 0.714 | 0.736 | 48 | 0.054 | 0.048 | 0.712 | 0.732 | 49 | 0.056 | 0.050 | 0.714 | 0.733 |
| | 800 | 46 | 0.055 | 0.049 | 0.731 | 0.757 | 46 | 0.052 | 0.047 | 0.722 | 0.750 | 47 | 0.057 | 0.050 | 0.719 | 0.746 |

Empirical type I error rate ($\alpha_\pi$ and $\alpha_{TS}$) and power ($1 - \beta_\pi$ and $1 - \beta_{TS}$) for the new test and for the classical two-sample log-rank test, respectively, under proportional hazards alternatives for Weibull distributed survival times with shape parameter $\kappa$ and 1-year survival rate $S_1 = 0.5$ in the control arm. Theoretical two-sided significance level: 5%. Underlying sample size of the historical group $n_A$ and the intervention $n_B$ calculated to achieve a theoretical power of 80% under the planning alternative $H_1$: $\Lambda_B = 2/3 \cdot \Lambda_A$ for the new test statistic using the sample size methodology presented in the previous section. Beside administrative censoring after $f_A = f_B = 3$ years in the historical and intervention group, we also introduced random Exponential distributed censoring with rates $\lambda_{C_A} = 0.2$ and $\lambda_{C_B}$ in the historical and intervention group respectively.

inflation depends in particular on the relative size of the control cohort compared to the intervention cohort [2].

Here we study and propose a non-parametric survival test that explicitly accounts for sampling variability of the reference curve. Our proposed test does not require the availability of complete data of the historical group. Instead, only the sample size and the graph of the Nelson-Aalen or Kaplan-Meier estimate is required to implement the test. Thus, in contrast to the two-sample log-rank test, our new test can also be calculated if access to the raw data of the historical group is restricted e.g. due to data protection precautions. However, when the full historical data is accessible, the new test may also be interpreted as two-sample test for survival distributions, while inheriting the interpretability from the underlying one-sample log-rank test. Admittedly, our simulations confirm that the two-sample log rank test is the method of choice to compare the data of a historical control cohort with the new data in single-arm Phase II trials if sampling variability of the reference curve is pronounced and if full data of the historical cohort is available.

We provided sample size calculation methodology for our new proposed test methodology, which accounts for the added variance of the estimation process for the the historical cohort.

As expected, the calculated sample sizes are of similar size as the sample sizes obtained through two-sample planning methodology like the Schoenfeld formula [9].

Traditionally, one-sample methods are popular because they require a small sample size: By using a one-sample log-rank test, not only the sample size of the control arm is saved, but also the reference curve is considered deterministic, thereby diminishing the variability of the test statistic. This again relevantly reduces the required sample size. However, if the reference curve itself is also estimated from data the assumption of a deterministic reference curve does not hold. Then the latter reduction of sample size comes at the price of an inflated type I error rate if not addressed by adequate statical considerations [2]. To obtain a valid test procedure the variability has to be accounted for, as our new proposed survival test does.

In summary we advise to use our new test methodology if the reference curve is obtained through an estimation process and access to the full historic data is restricted. It is important to note, that the usual use case of the two-sample log-rank test is a randomised controlled trial which fundamentally differs from the case of restricted reference data which is by design unrandomised and leaves no room for adjustment techniques of confounders since individual patient level data are not available. Therefore it is crucial to exercise extreme caution when using a historical control from a summarized Kaplan Meier survival curve. Thus application of the new test may lead to significant differences between the compared survival curves, but it may be unclear whether this difference stems from treatment or confounding. For this reason the methodology is suitable for phase II trials but not for phase III trials as a basis for drug approval or health technology assessment decision making. In addition, when choosing the reference curve, it must be ensured that the recruitment and censoring mechanisms of the historical cohort are equal to those of the new cohort in order to guarantee an exact type I error rate control. Therefore, we advise to use the classical two-sample log rank test in case of individual patient level survival data availability, since it does not rely on the assumption of equal recruitment and censoring mechanisms. Nevertheless, the new test has a similar performance as the two-sample log rank test which is known to be optimal under proportional hazards. Furthermore it should be noted that the new test methodology can be used with existing standard software, as the approach is equivalent to an adjustment of the nominal $\alpha$-level in the standard one-sample log-rank test.

## Supporting information

**S1 Appendix. Mathematical details.** Mathematical statements and corresponding proofs.
(PDF)

**S1 Text. Sample size Pi.** R code. Supplementary R code for sample size calculation.
(TXT)

**S2 Text. Simulation function.** R code. Supplementary R code for simulating design scenarios.
(TXT)

**S3 Text. Cpp code.** R code. Supplementary R code for simulating design scenarios.
(TXT)

**S4 Text. Simulation different censoring.** R code. Supplementary R code for simulations for tables with differing censoring mechanisms.
(TXT)

**S5 Text. Simulation different followUp.** R code. Supplementary R code for simulations for tables with differing follow-up lengths.
(TXT)

## Acknowledgments

We would like to thank Prof. Dr. med. Holger Reinecke for revising our real world example. We would like to thank our cooperation partners at the AOK Research Institute (WIdO) for granting us permission to use their data within our real world example. This permission was granted within the framework of the GenderVasc project (Gender-specific real care situation of patients with arteriosclerotic cardiovascular diseases). We acknowledge support from the Open Access Publication Fund of the University of Muenster.

## Author Contributions

**Conceptualization:** Jannik Feld, Anastasia Janina Hobbach, Rene Schmidt.

**Formal analysis:** Jannik Feld, Moritz Fabian Danzer, Rene Schmidt.

**Funding acquisition:** Rene Schmidt.

**Investigation:** Jannik Feld, Moritz Fabian Danzer, Rene Schmidt.

**Methodology:** Jannik Feld, Moritz Fabian Danzer, Rene Schmidt.

**Project administration:** Jannik Feld, Andreas Faldum, Rene Schmidt.

**Resources:** Jannik Feld, Andreas Faldum.

**Software:** Jannik Feld.

**Supervision:** Andreas Faldum, Rene Schmidt.

**Validation:** Jannik Feld, Moritz Fabian Danzer, Rene Schmidt.

**Visualization:** Jannik Feld.

**Writing – original draft:** Jannik Feld, Rene Schmidt.

**Writing – review & editing:** Anastasia Janina Hobbach.

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
