## [Decision Letter · Decision Letter 0]

12 Dec 2023

PONE-D-23-29714On historically controlled survival trialsPLOS ONE

Dear Dr. Feld,

Thank you for submitting your manuscript to PLOS ONE. After careful consideration, we feel that it has merit but does not fully meet PLOS ONE’s publication criteria as it currently stands. Therefore, we invite you to submit a revised version of the manuscript that addresses the points raised during the review process.

I agree with the comments of the reviewers and have some additional remarks and recommendations. Besides, the comments of the reviewers, the following issues need attention:

(1) The title should be changed to describe the content of the paper (see suggestions of the reviewers).

(2) The differences of the new (corrected one-sample) test and the usual (two-sample) log-rank test should be worked out more thoroughly. Which information from the historical reference curve cannot be extracted to apply the usual (two-sample) log-rank test?

(3) The repetitions from earlier papers should be reduced.

(4) The information that sample-size formulas are propsed and a simulation study was performed should be added to the Abstract

(5) An example should be added illustrating the application of the new method compared to the usual one-sample log-rank and two-sample log-rank tests.

(6) The new test called corrected one-sample log-rank test can also be viewed as new two-sample test for the case that complete historical survival data on patient level are not available. However, with this interpretation another important difference between the two data situations is that the classical two-sample log-rank test is usually applied in RCTs, whereas the new test is applied in a non-randomised data situation. The increased uncertainty due to missing randomisation in applications of the new test should be discussed at the end of the paper.

We look forward to receiving your revised manuscript.

Best wishes,

Ralf

Ralf Bender, Ph.D.

Academic Editor

PLOS ONE

Journal Requirements:

The work of Moritz Fabian Danzer was funded by the German Science Foundation 

(Deutsche Forschungsgemeinschaft, DFG, grant number 413730122).

The work of Moritz Fabian Danzer was funded by the German Science Foundation

(Deutsche Forschungsgemeinschaft, DFG, grant number 413730122).

The work of Moritz Fabian Danzer was funded by the German Science Foundation 

(Deutsche Forschungsgemeinschaft, DFG, grant number 413730122).

4. We are unable to open your Supporting Information file Simulation_Function.R, Sample_size_Pi.R, Simulation_Different_Censoring.R and Simulation_Different_FollowUp.R. Please kindly revise as necessary and re-upload.

Reviewers' comments:

Reviewer's Responses to Questions

**Comments to the Author**

1. Is the manuscript technically sound, and do the data support the conclusions?

Reviewer #1: Partly

Reviewer #2: Partly

2. Has the statistical analysis been performed appropriately and rigorously? 

Reviewer #1: Yes

Reviewer #2: Yes

3. Have the authors made all data underlying the findings in their manuscript fully available?

Reviewer #1: Yes

Reviewer #2: Yes

4. Is the manuscript presented in an intelligible fashion and written in standard English?

Reviewer #1: Yes

Reviewer #2: Yes

5. Review Comments to the Author

Reviewer #1: (identical review attached as a file)

Review of PLOS ONE manuscript PONE-D-23-29714

„On historically controlled survival trials“

Overall, this is an interesting, well-written and relevant manuscript.

General comments

1. I think the title of the manuscript should be more specific. It should include „log-rank test“.

2. To demonstrate practical relevance, I suggest that the manuscript might benefit from a brief additional paragraph on a real world example,showing how the conclusion (the p-value) from a specific phase II trial would change, had the new test been applied.

3. The description of the censoring scheme assumed in the sample size calculation and the simulation study is unclear (see specific comments below). Is it recruitment period plus fixed follow-up or recruitment period plus minimum follow-up until end of trial? Both is found in practice trials with survival endpoints.

4. While it is true that the new test can be implemented when only the historic sample size and the Nelson-Aalen (or 1 minus Kaplan-Meier at maximum follow-up) of the control group is known, I wonder how relevant and accessible the assumption of identical follow-up schemes is in practice. This aspect should receive some more room in the discussion.

Specific comments – being a non-native speaker myself, I may be wrong with the corrections that I propose concerning the English language.

• Abstract, It thus applies in settings, when -> It thus applies in settings when (omit German comma)

• L. 17 This may be for a variety reasons -> This may occur for a variety of reasons

• L. 20 an survival distribution estimator -> a survival distribution estimator

• L. 27 Please rearrange references in numerical order of citation

• L. 30 an estimator of historical survival curves -> Should this read „of the historical survival curve“ (singular, there is only one control group)?

• L. 51/52: Please add „for i in N_x“, as in the sum in line 61.

• L. 53 In addition, and also as usual, one would assume that the data are i.i.d. samples from groups A and B. You require identical distributions in L. 69, but not independence.

• Equn (2): I suppose N_i(s) should read N_x,i(s)?

• L. 64 Please add „for s greater-equal 0“

• L. 65 ff. The „d“ in integrals would preferrably not be set in italics, for clarity.

• L.79, please introduce s_max

• L. 95.5 (equation): Please introduce X_B,i

• M_0-hat is defined twice, in L.101 and equation (5)

• L. 115, stochastic processes is -> stochastic process is

• L. 116, it was shown, that -> it was shown that (omit German comma)

• L. 121.5, leads to a under-estimation -> leads to an under-estimation

• L. 128 and ff, Notice, that -> Notice that (omit German comma)

• L. 134 or equivalent the Kaplan Meier estimator -> or equivalently the Kaplan-Meier estimator

• L. 138.5 You need to introduce Λ_x, probably as Λ_x(s_max)

• L. 140, please introduce a_B, f_B, r_B. You might use terms like recruitment period and fixed-time or additional follow-up.

• L.151,it’s not quite clear where you assume the plateau, in particular since f_B has not been introduced - I don’t understand ‚at the edge of the observation window‘ – does that mean AT s_max? If the plateau occurs during the interval between a_B and a_B+f_B, then (ii) would equal S(a_B) not S(f_B). Please clarify. Also, I’m not sure whether „No censoring despite administrative censoring“ means what I would call „No censoring except for administrative censoring“

• L. 154 declines short after the procedure -> declines shortly after the procedure

• L. 201, independent exponential distributed censoring -> independent exponentially distributed censoring

• L. 208, please indicate also the one-year survival rate in the intervention group in this scenario (S_T_B(1)).

• L 229, scenarios with particular small sample sizes -> scenarios with particularly small sample sizes

• L 270, we advice to use our new test -> we advise to use our new test

• Legend to table 1, „Underlying sample size of the historical group n_A and the intervention n_B calculated to achieve a theoretical power“: I think n_A should be viewed as given historically, while n_B is calculated to achieve a theoretical power. Please rephrase.

Reviewer #2: The present paper continues a previous work of the authors on one-sample testing and the question of the reference curve, as well as comparisons with the two-sample test.

I found the paper generally well-written and it addresses an interesting subject. Yet, I have a few main concerns, next to a number of additional remarks.

Main concerns:

1. It should be made clearer what are the differences from the previous paper (Ref. 12). As far as I can tell, the first sections (until but excluding the section on Sample-size calculation) are repetitions of content from Ref. 6. Also, it should be made more explicit in the section on Sample-size calculation which results are known from the literature (e.g., Ref. 4) an which results are new. Having said this, I am wondering whether the new content is sufficient for justification of a publication in PLOS ONE. But this is not for me to decide. Either way, I believe the paper could benefit from a more specific title (and abstract) which highlights the new contributions (and an abstract which distinguishes a bit better from previous works). One candidate title could be: "Sample size planning for the one-sample log-rank test with a historical control."

2. What is (/are) the real difference(s) between the proposed adjusted "one-sample" test and the two-sample test? (Also the simulation results are very similar for both kinds of tests.) To my understanding, all the relevant data from the historical reference sample can be extracted from the historical reference curve, so it seems like a classical two-sample problem to me. What is more, the two-sample test is known to have a test statistic which is approximately N(0,1)-distributed under the null hypothesis. To my understanding, the authors suggested a one-sample variant which uses an alternative variance estimator to account for the factual two-sample problem. Now, if the proposed test statistic is also approximately N(0,1)-distributed, does not imply that the proposed test approximates the classical two-sample test? It should be made clearer what are the actual differences between both tests. For example, will the difference of the variance estimators decrease to 0 rapidly? I think a good understanding of this is of the essence to underline the relevance of this paper.

3. p.4, formula (4): This is not the Greenwood-type variance estimator which is not a problem per se because the existence of hazard rates is assumed, i.e., no ties in the data. However, a relevant fraction or majority of data sets include ties, for which the estimator (4) would result in a biased estimator of the variance. In order to make the paper more relevant and more generally applicable, I suggest to switch to the Greenwood-type estimator. Correspondingly, it would be interesting to see whether the simulation results are affected. (My guess is: probably not much because continuous distributions were used there.) The reason why I am also bringing this up: the historical curves typically also result from data sets that include ties. So, to have a variance estimator that works for the historical data set, one should (also) use the Greenwood-type estimator for it.

Additional major comments:

I. p.5, ll.121-122: I find it surprising that the factor R only depends on the sample sizes. Even under the null hypothesis, I would be surprised. Is it only valid under the condition of equal samplingschemes (and censoring) in both groups? I think some clarification is needed here.

II. p.5, ll.128-135 I fail to see the real difference between Z and Z_pi. Is the gist that R is only approximately equal to the reciprocal of sqrt(1+pi)? Does it still matter for big sample sizes? Or does it have to do with my major comment I.?

III. p.5, ll.134-135: I don't let this count; in my opinion, knowing the estimated reference curve is almost as good as knowing the historical data set. From the reference curve, one could also retrieve an estimate of sigma_A.

IV. p.6, l.137: what is the motivation/justification for the definition of d at the end p.5, and for the calculation of d at the beginning of p.6?

Additional remarks and questions:

- p.2, l.56-57: Why does group membership play a role in calendar time but not in study time? This should be explained, and the readability of the sentence could also be improved. Also, another part to correct: "are not simulatenous".

- p.3: The definition of J_x seems to be missing.

- p.3: About H_real: this is a rather strange formulation. Please point out that this is not a real null hypothesis because it depends on a random object. (Otherwise readers might get the impression it is acceptable to formulate hypothesis in this way.)

- End of p.3 = end of the section General aspects: This might be a good spot for explaining the need for a test statistic which is different from the classical two-sample test statistic.

- p.4, l.117: I guess typical notation for the variance estimator would involve the power 2 (for \\hat \\sigma_A).

- p.6, l.137: in the definition of omega, is the ratio of the cumulative hazards to be assumed time-constant? This would be a strong restriction in my opinion.

- p.6, l.140: What are a_B, f_B, r_B supposed to be? (Knowing this will also facilitate the understanding of (i) and (ii) in l.151.)

- p.7, ll.164-165: there is a index mistake (I believe) in the second display: "c1".

- p.7, ll.176-177: "our test with consideration of reference

curve variability, may also be interpreted as a two-sample survival test." Knowing more precisely what is the difference between the proposed novel test and the two-sample test would help understanding this sentence. (Or perhaps reformulate.)

- p.8, l.206: instead of S_1, it should probably be s on the right-hand side.

- p.9, ll.249-251: Again, it might not be easy for readers to grasp the essence of this sentence.

- p.9, ll.257-259: Instead of "interestingly", was this not rather to be expected?

- p.10, ll.272-273: Again, this sentence might increase confusion. If it has been assumed (implicitly?) that the recruitment and censoring mechanism are similar, this should be made explicit in the paper.

- References: Did the authors specify the wrong journal for Ref. 6? I found this paper in Statistics in Biopharmaceutical Research.

6. PLOS authors have the option to publish the peer review history of their article (what does this mean?). If published, this will include your full peer review and any attached files.

Reviewer #1: **Yes: **Erika Graf

Reviewer #2: No

---

## [Author Response · Author response to Decision Letter 0]

7 Mar 2024

We answered the comments in detail within the "Response to Reviewers.docx" document.

---

## [Decision Letter · Decision Letter 1]

19 Apr 2024

PONE-D-23-29714R1Designing one-sample log-rank tests in the absence of historic individual patient dataPLOS ONE

Dear Dr. Feld,

Thank you for submitting your manuscript to PLOS ONE. After careful consideration, we feel that it has merit but does not fully meet PLOS ONE’s publication criteria as it currently stands. Therefore, we invite you to submit a revised version of the manuscript that addresses the points raised during the review process.I agree with the reviewers that the revision improved the manuscript considerably and that the authors addressed most of the points adequately. Only a few issues remained and should finally be solved. Besides the comments of the reviewers please consider the following 2 issues. 1) I also think that the term "corrected one-sample test" is misleading. In fact the new test it is a two-sample test, which does not require the full individual patient data in one of the 2 samples.2) In the discussion you correctly added the remark that "extreme caution" is required when historical control data are used because this leads to a non-randomized study setting. This point should be acknowledged more thoroughly. I recommend to add that the application of the new test in a non-randomized study setting means that the test can find a significant difference between the 2 compared survival curves, but that it remains unclear whether this difference is due to the treatment or due to confounding. The consequence is the test is suitable for e.g., phase II trials but not for phase III trials as basis for drug approval or HTA decisions.

We look forward to receiving your revised manuscript.

Best wishes,

Ralf

Ralf Bender, Ph.D.

Academic Editor

PLOS ONE

Journal Requirements:

Reviewers' comments:

Reviewer's Responses to Questions

**Comments to the Author**

1. If the authors have adequately addressed your comments raised in a previous round of review and you feel that this manuscript is now acceptable for publication, you may indicate that here to bypass the “Comments to the Author” section, enter your conflict of interest statement in the “Confidential to Editor” section, and submit your "Accept" recommendation.

Reviewer #1: (No Response)

Reviewer #2: All comments have been addressed

2. Is the manuscript technically sound, and do the data support the conclusions?

Reviewer #1: Yes

Reviewer #2: Yes

3. Has the statistical analysis been performed appropriately and rigorously? 

Reviewer #1: Yes

Reviewer #2: Yes

4. Have the authors made all data underlying the findings in their manuscript fully available?

Reviewer #1: Yes

Reviewer #2: No

5. Is the manuscript presented in an intelligible fashion and written in standard English?

Reviewer #1: Yes

Reviewer #2: Yes

6. Review Comments to the Author

Reviewer #1: The revision has strengthened the manuscript. The authors addressed the points raised adequately. Here are some remaining issues which could further strengthen the text.

Upon re-reading the sentence "Here, we propose a corrected one-sample log-rank test that allows to account for the sampling error of the

reference curve without knowledge of the full underlying historical survival time data.", I actually asked myself whether'one-sample' log-rank test is a misnamer for the corrected test procedure. The corrected log-rank test compares a historical with a new data sample, the relevant aspect of the corrected procedure being that, compared to the usual two-sample log-rank test, summary statistics rather than individual patient data are being used for the reference arm. In fact, the very point you make in favour of the corrected procedure is random variability of the inherent assumption for the control arm. Should this indeed be called a one-sample test?

Introduction of the example has strengthened the manuscript and serves the technical purpose. The description of the background given previous paradoxical results is also interesting. We don't need to go deep into a subject-matter discussion. However, please consider adding a brief sentence saying that these paradoxical results were not confirmed in your analysis, since the Kaplan Meier curve of the obese is below that of the non-obese.

L. 72f. "We denote by a_B the recruitment period length, with f_B the follow up period length resulting in a trial period length of a_B + f_B for the new intervention trial." Please clarify still further. Is every patient followed up for a period of length only f_B, or is f_B the minimum, a_B + f_B the maximum duration of follow up for the last and first recruited patient, respectively (all patients followed up until calendar time end of trial)? (The latter seems to be the case, according to L. 174ff)

L. 307 "Conceptually, the proposed new test also sheds light on a general strategy for lifting existing methodology for single-arm survival trials to a randomized, multi-arm setting" - I find this a somewhat strange statement (can a test shed light on a strategy?). I would rather say that by taking into account the control group variability, your methodology integrates an element into single-armed comparisons with historical control groups that is naturally present in randomized trials, although obviously it still cannot address the potential for selection bias due to the lack of randomization.

Minor remarks

L. 56 from a group of historical patients, which were treated under standard -> historical patients who (omit German comma)

L. 75 Please introduce the recruitment rate r_B by an equation (presumably n_B/a_B?)

L.96 Φ−1 is the standard normal: Please move to L. 91.5

L. 99 one-sample log-rank test could be used to test H0, if the reference cumulative hazard function Λ_A was known -> test H0 if ... were known (omit German comma)

LL. 132,133: german -> German

LL. 134, 138.5, 146.5: an myocardial -> a myocardial

L. 165: is lower then -> is lower than

L. 173, for all s ≥ s: Please correct

L. 173.5, ... pi. This number of events...: Please reformulate e.g. as '...pi. The number d of events...' (I was confused here because you seemed to describe pi as the number of events).

L. 224 based on the assumption, that: omit German comma

L. 255 "In this case, the breadth of a 95%-confidence interval for an underlying true value of 0.05 is about 0.004". I don't understand this sentence (although I have a guess), please rephrase. True value of what? What confidence interval? None is shown in the tables.

L. 304 f. "Our methodology closes this loophole of artificially reduced sample sizes at the prize of inflated type I error rate." Please rephrase for clarity. A prize has to be paid for the reduced sample size - the sentence could be misread as: for closing the loophole.

Reviewer #2: Thank you for addressing my previous comments.

I only have a minor comment left:

- p. 6, at the end, it currently reads "for all s>=s".

7. PLOS authors have the option to publish the peer review history of their article (what does this mean?). If published, this will include your full peer review and any attached files.

Reviewer #1: **Yes: **Erika Graf

Reviewer #2: No

---

## [Author Response · Author response to Decision Letter 1]

22 May 2024

We answered the comments in detail within the "Response to Reviewers.docx"

document.

---

## [Editor Report · Decision Letter 2]

28 May 2024

Two-sample survival tests based on control arm summary statistics

PONE-D-23-29714R2

Dear Jannik,

We’re pleased to inform you that your manuscript has been judged scientifically suitable for publication and will be formally accepted for publication once it meets all outstanding technical requirements.

Best wishes,

Ralf

Ralf Bender, Ph.D.

Academic Editor

PLOS ONE

Additional Editor Comments (optional):

Line 320: I think grammatically correct is "data are not available" (rather than "data is not available"); I recommend to correct this (at the latest when you correct the proofs).

---

## [Editor Report · Acceptance letter]

5 Jun 2024

PONE-D-23-29714R2 

PLOS ONE

Dear Dr. Feld, 

I'm pleased to inform you that your manuscript has been deemed suitable for publication in PLOS ONE. Congratulations! Your manuscript is now being handed over to our production team.

Kind regards, 

on behalf of

Professor Ralf Bender 

Academic Editor

PLOS ONE